# Data-Centric and Model-Centric AI: Twin Drivers of Compact and Robust Industry 4.0 Solutions

Oussama H. Hamid

Faculty of Computer Information Sys, Higher Colleges of Technology,
Abu Dhabi P.O. Box 41012, United Arab Emirates; ohamid@hct.ac.ae

**Abstract:** Despite its dominance over the past three decades, model-centric AI has recently come under heavy criticism in favor of data-centric AI. Indeed, both promise to improve the performance of AI systems, yet with converse points of focus. While the former successively upgrades a devised model (algorithm/code), holding the amount and type of data used in model training fixed, the latter enhances the quality of deployed data continuously, paying less attention to further model upgrades. Rather than favoring either of the two approaches, this paper reconciles data-centric AI with model-centric AI. In so doing, we connect current AI to the field of cybersecurity and natural language inference, and through the phenomena of 'adversarial samples' and 'hypothesis-only biases', respectively, showcase the limitations of model-centric AI in terms of algorithmic stability and robustness. Further, we argue that overcoming the alleged limitations of model-centric AI may well require paying extra attention to the alternative data-centric approach. However, this should not result in reducing interest in model-centric AI. Our position is supported by the notion that successful 'problem solving' requires considering both the way we act upon things (algorithm) as well as harnessing the knowledge derived from data of their states and properties.

**Keywords:** artificial intelligence; adversarial samples; current AI; data-centric AI; deep learning; Industry 4.0; machine learning; model-centric AI

## 1. Introduction

With the fourth industrial revolution, also referred to as Industry 4.0, a new wave of automation set off carrying novel industrial applications that promise to redefine the interaction between man and machine [1]. Businesses are increasingly shifting their service models from traditional automation, in which the integration of software systems on service platforms mainly depends on the outstanding programming skills of human software developers and the deployment of appropriate application programming interfaces (APIs) [2,3], to engineered systems that integrate sensing abilities, computational prowess, control, and network utilities into *cyber–physical* objects that can connect to each other over the 'Internet of Things' (IoT) [4].

Data are a conspicuous hallmark of Industry 4.0 technologies [5]. Examples range from cloud computing [6], where large volumes of data can be stored, analysed, and processed more efficiently and cost-effectively with the cloud, over edge computing [7], which provides low end-to-end latency in real-time production operations [8] by allowing efficient near-sensor data analysis [9], to digital twins [10], where simulations of multiple systems' processes can be run on virtual environments, pulling data from IoT sensors and whatever objects connected to the Internet [11].

Of particular importance in moving data around the grid are the fields of artificial intelligence (AI) and machine learning (ML) [12]. Together, they can facilitate the creation of sustainable solutions and scalable business models by taking full advantage of available data volumes in a way that exceeds the use of the data generated within the concerned business field [13,14] to include the data obtained from other fields and businesses [15].

At its core, an AI system is mainly an algorithm (code) that solves a problem by learning prototypical features (patterns) from large volumes of data. Data could be in any of different forms (for example, text, audio, image, and/or video). ML is a subfield of AI that characterises the system's capacity to spot patterns that are otherwise undetectable by humans [16]. It achieves this by using general-purpose procedures, which enable the AI system to solve the problem at hand without being explicitly programmed to do so [17]. Yet, for this to occur, the system needs to process data in a way that facilitates interpretation and provides context [16]. Typically, this is made possible by letting human experts label the data as part of the data preparation process, which includes collecting, curating, cleaning, and transforming raw data before processing and analysing it. This subtask is known as 'data annotation'.

Currently, there are two different views within the ML community on how to improve the performance of AI systems: model-centric AI and data-centric AI. In model-centric AI, developers of an AI system successively upgrade a devised model (algorithm/code), while holding the volume and type of data collected fixed. Conversely, one can hold the model fixed and continuously improve the quality of data until reaching a high level of overall performance in terms of dealing with noise in the data. This is the data-centric AI approach (Figure 1). Due to the fact that the training processes in both approaches run in a continuous, iterative fashion, some refer to them as 'model-cycle' and 'data-cycle' approaches, respectively, [18].

Over the past three decades, model-centric AI was dominantly used in both research and industry. In research, for example, more than 90% of the published AI research projects utilised a model-centric approach [19]. Recently, however, voices are getting louder in the promotion of data-centric AI. Reasons for this change of heart range from the lack of sufficiently large datasets [20] to the need for highly customised solutions [21].

The concept was introduced by Andrew Ng, founder of 'DeepLearning.AI' (an educational platform) and Adjunct Professor at Stanford University. In a live stream presentation on 24 March 2021, Ng showcased the use of a data-centric approach in the detection of foreign-particle defects in steel sheets, which resulted in a considerably higher performance of the baseline model of an AI system devised to detect steel defects, compared with the almost no-improvement outcome when a model-centric approach was applied to the baseline model [19]. Furthermore, Ng showed that the application of the data-centric approach to other tasks such as solar defect detection and surface inspection confirmed its superiority over the model-centric approach in both model accuracy and speed of learning [19]. Strikingly, when asked whether to improve the code or the data in order to enhance the accuracy of the devised model in the steel inspection problem, the majority ($\approx$80%) of approximately 2000 voters in the live stream presentation chose improving the data.

Rather than the typical 'either/or' approach, this work supports a 'both/and' one. Though a study aiming at illustrating the relative potency of data-centric AI would benefit from experimental procedures, in which data-centric AI is tested against its model-centric counterpart. Doing this, however, is beyond the scope of our goal planning for the current study. Specifically, within the AI/ML community, we observe a kind of 'paradigm shift' in the way AI/ML models are devised. Within this constellation, more and more researchers are calling for replacing the model-centric approach with a data-centric alternative. They support their position with experimental findings from the applications they work on. That goes well when the aim is to show the potency of the new approach (i.e., the data-centric AI). Yet, in times of paradigm shifts, success stories from isolated experimental research works provide no proof supporting the supremacy of one approach over the other alternatives. Rather, one should show the supremacy of the supported approach (data-centric AI) over other alternatives (model-centric AI) through experimental results in every possible application. However, no research study can afford this.

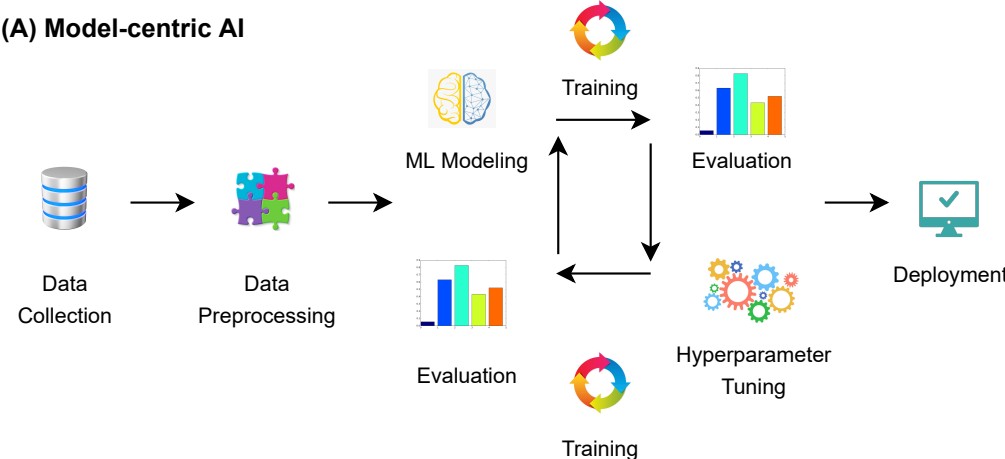

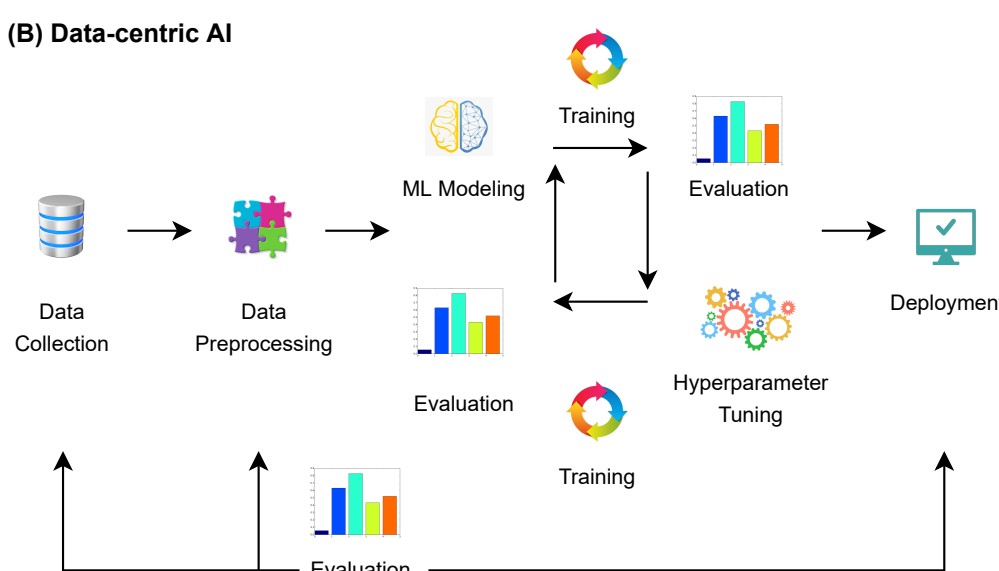

**Figure 1.** Cycles of model-centric AI and data-centric AI (source: [17]).

Indeed, what advocates of data-centric AI are demonstrating through their experimental findings is merely the limitation of model-centric AI. Though we share this view with them, our central argument in the current work remains that rather than championing any of the two alternatives, we opt for a 'both/and' approach. We argue that overcoming the limitation of model-centric AI does, indeed, require paying extra attention to the alternative data-centric AI approach. However, this should not result in reduced interest in the model-centric approach, for it provides us with the anchor necessary to assess the quality of an AI system over the course of upgrading the dataset.

*Our Contribution*

This paper has been significantly extended from previous work [17]. In the current work, we use a comparative analysis methodology, where data-centric AI and model-centic AI are compared and contrasted (Table 1). As elements of the compare-and-contrast approach, the paper takes Andrew Ng's live stream presentation on 24 March 2021 [19] as frame of reference. Moreover, grounds for comparison are given through the increasing number of works by other researchers, who suggest to shift from model-centric AI to data-centric AI. The main thesis in this work is: "whereas model-centric AI suffers from

performance limitations, developing both approaches in a complimentary interplay would benefit current AI more than focusing on improving datasets only (albeit important). The main contributions of the present paper are as follows:

1.  We compactly review and discuss the deep learning technique, highlighting its role in driving current AI hype (Section 2.1);
2.  We connect current AI to the fields of cybersecurity and natural language inference, and through the phenomena of 'adversarial samples' and 'hypothesis-only biases,' respectively, showcase the limitations of model-centric AI in terms of algorithmic stability and robustness (Sections 3.2 and 3.3);
3.  We further motivate a data-centric AI approach by elucidating the effect of the ongoing growth of the IoT, supporting our approach with the latest relevant data (Section 4);
4.  We reconcile data-centric AI with model-centric AI, providing further arguments for the 'both/and' view instead of the evidently suboptimal alternative of the 'either/or' perspective (Section 6).

## 2. Related Work

### 2.1. Deep Learning: A Model-Centric Key Driver of Current AI

Historically, John McCarthy (1927–2011) was the first to coin the term 'Artificial Intelligence' in 1956. He defined it as the *"science and engineering of making intelligent machines, especially intelligent computer programmes"* [22]. Today, much of the hype around AI is attributed to a particular technique called deep learning (DL) [23] which is a modified version of artificial neural networks (ANNs) (Figure 2).

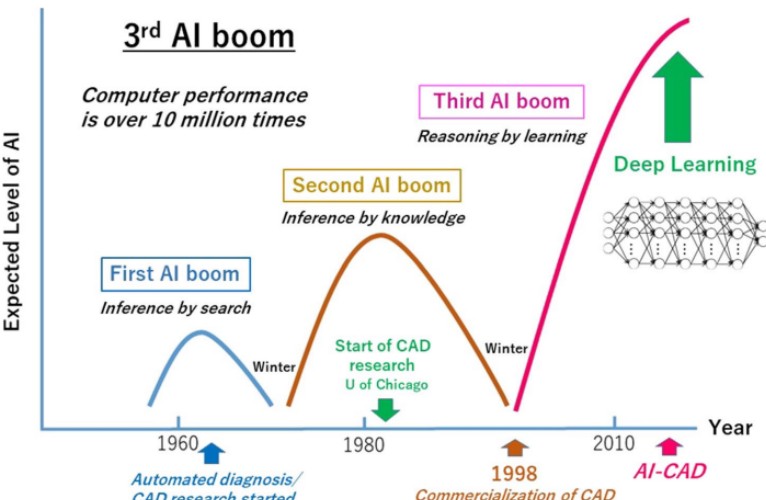

**Figure 2.** Current AI. The 3rd AI hype, in which previous computer-aided detection/diagnosis for medical images is augmented by deep learning, is preceded by two cycles of AI hype; source: [24].

In an accelerated pace of enhancing AI/ML prowess, DL was repeatedly reported to have accomplished human-level performance in a variety of applications. Examples range from object classification [25], over beating world-class players in Go and Poker [26,27], to detecting cancer from X-ray scans [28], and translating text across languages [29]. Although the theoretical foundations of DL were laid in the early 1940s, it was not until 2012 that researchers from AI/ML community and other related disciplines celebrated it as a technique most mimicking the human brain. This occurred when Geoffery Hinton, a British Canadian cognitive psychologist and computer scientist, along with two of his students, won the annual ImageNet contest, in its third version, with remarkably higher performance and accuracy than previous state-of-the-art algorithms [30].

### 2.1.1. ANN: Learning by Adjusting Weights

In a typical artificial neural network (ANN), learning occurs by adjusting the 'weights', $w$, that amplify or damp signals, $x$, carried by each connection between any two nodes in the ANN (Figure 3). A number of hidden layers, each constituting a module that is roughly analogous to some processing centres in the human brain, are used to backpropagate gradients. At each layer, we first compute the total input $z = \sum wx$ to each unit, which is a weighted sum of the outputs of the units in the layer before. Then, a nonlinear function $f(.)$ is applied to $z$ to obtain the output of the unit. For simplicity, bias terms were omitted. The nonlinear functions used in artificial neural networks include the rectified linear unit (ReLU) $f(z) = \max(0, z)$ along with the conventional sigmoids, such as the hyperbolic tangent, $f(z) = (\exp(z) - \exp(-z))/(\exp(z) + \exp(-z))$ and the logistic function, $f(z) = 1/(1 + \exp(-z))$ [31].

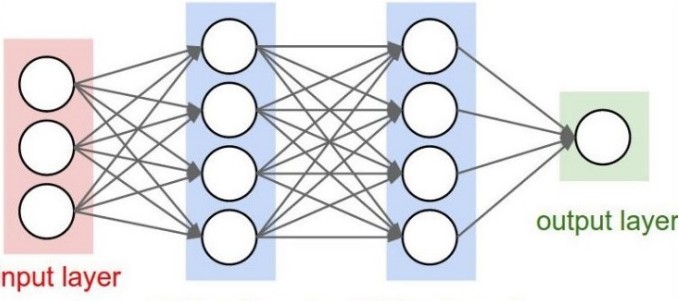

**Figure 3.** Artificial neural network with two hidden layers (schematic).

### 2.1.2. Learning over Multiple Levels of Abstraction

Compared with traditional ANNs, a DL network learns data representations over multiple levels of abstraction [31]. This is achieved by deploying significantly more layers of simple, nonlinear modules (neurons) that transform the internal representation of certain aspects of input data in one layer into another internal representation at a higher level of abstraction. A backpropagation algorithm works its way back down through the layers, fine-tuning the weights of the neural network in proportion to how much each individual weight contributes to the overall error.

Usually, the number of layers in a DL network ranges from 5 to 20 layers, hence the term 'deep'. Today, however, neural network models in commercial applications often use over 100 layers.

Different from classical ML techniques, which require careful engineering and considerable domain expertise (by human experts) in order to extract relevant aspects of input data into a suitable internal representation (also known as feature vector), DL models can operate without the need for any explicit human intervention in the feature extraction step. Specifically, a DL algorithm learns features from data implicitly by using general-purpose procedures. Indeed, it is due to this capacity that DL was linked to the way the human brain learns to solve problems. Importantly, it has been proved that with a non-polynomial activation function, any continuous function can be approximated to any degree of accuracy [32], which makes ANN mathematically equivalent to universal computers, a result known as the universal approximation theorem.

### 2.2. Data: Sustainable Fuel of Current AI

Several factors explain why just in the past decade DL could evolve into what it practically became and achieved a lot of what AI researchers hoped to see in the early days of developing the technology. Most importantly, DL works well with sufficiently large volumes of data. Such data stem from various sources in the 'Global Datasphere' and is continuously alternating in textual, visual, and acoustic forms. The Global Datasphere consists of cloud data centres,

enterprise infrastructure such as cell towers and branch offices, and endpoints such as PCs, smartphones, sensors, social media, wearable devices, etc.

DL algorithms need data to train a model that can check statistical similarities between the currently considered instances and previously probed ones, so as to uncover hidden patterns (unsupervised learning) and make future predictions (supervised learning) about unseen instances [33]. The more granular, voluminous, and diverse the used data, the higher the performance and accuracy of the underlying algorithm. In the past, neither data nor the infrastructure required for storage and data transfer were available and mature as they are today.

According to International Data Corporation (IDC), a market intelligence company specialised in measuring created, consumed, and stored data each year, Global Datasphere would continue to expand almost exponentially (Figure 4). For example, while the volume of data created and replicated in 2010 did not exceed a lean value of 1 zettabyte (ZB), it reached a value of 64.2 ZB in 2020 [34]. It is expected that more than 180 ZB will be created in 2025 (1 ZB = $10^{12}$ gigabytes).

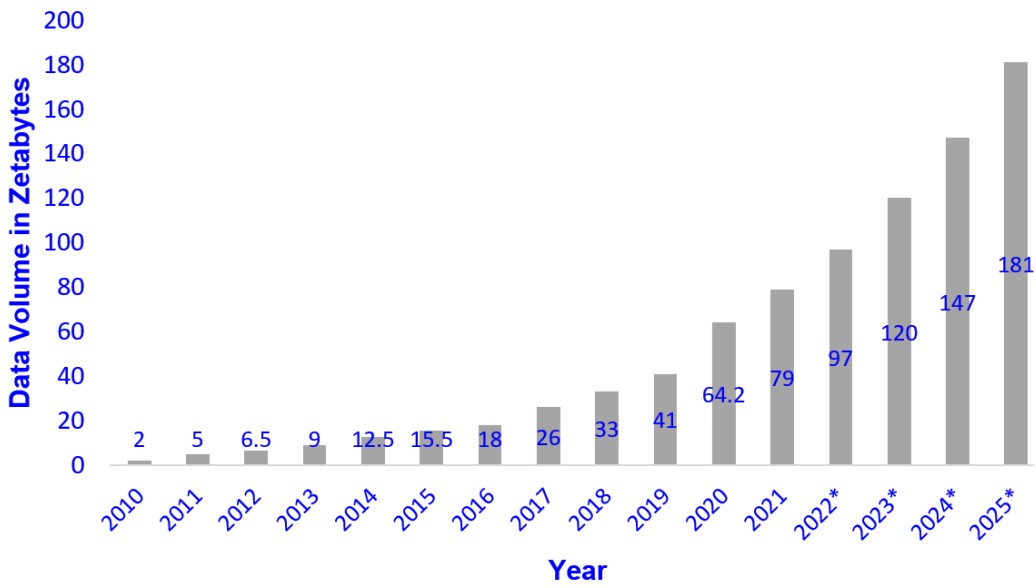

**Figure 4.** Annual size of data creation, consumption, and storage according to IDC. The stars denote forecast values. Data adapted from [35] with reference to [36].

## 3. Limitations of Model-Centric AI

Model-centric AI assumes ML solutions that mainly focus on optimising model architectures (algorithm/code) along with the underlying hyperparameters. Data within this approach are created almost only once and are kept the same over the life cycle of the AI system's development. Despite numerous success stories, model-centric AI has been placed under considerable strain. On top of it comes a narrow scope of business applicability along with the inherent vulnerability of DL/ML techniques to adversarial samples (also known as adversarial examples). The following subsections elaborate on this.

### 3.1. Narrow Business Applicability

The fact that training a DL algorithm requires utilising huge volumes of data causes the underlying model to suffer from an irreparable limitation [37–39]. Specifically, it works well in businesses and industries where consumer platforms with hundreds of millions of users can freely rely on generalised solutions. In such settings, a single AI system would satisfy the majority of users, while outliers would be negligibly ineffective. Examples for such businesses and industries include the advertising industry, where companies

such as Google, Baidu, Amazon, and Facebook possess mounds of data (not rarely in a standardised format), which they can deploy in creating their model-centric AI systems.

Industries such as manufacturing, agriculture, and healthcare, which require tailored solutions rather than one-size-fits-all recipes, cannot be served by standardised solutions similar to the ones provided by a single AI system. Instead, they should conceptualise their approach so as to ensure their model (algorithm) learns what it should learn by having comprehensive data that cover all important cases and are labeled consistently.

### 3.2. Vulnerability to Adversarial Samples

Adversarial samples are instances with small, deliberate feature perturbations that cause a DL/ML model (algorithm) to make false predictions [40]. Experts are particularly concerned about this phenomenon, for it questions the stability and robustness of DL networks with respect to noisy inputs [41]. To explain, consider a deep neural network algorithm (model) designed to perform object recognition tasks. If the algorithm can generalise well, then the network is supposed to be stable and/or robust to small perturbations to its input. This is because small disruptions in an input image would not change the categories of the objects appearing in that image. Surprisingly, though, when small, indiscernible perturbations are *deliberately* added to the test images of some objects, the network tends to change its predictions arbitrarily [41].

Attackers exploit this security gap by adding small, nonrandom, and imperceptible perturbations to benign samples—in some benchmark datasets, such as ImageNet [42], the differences between the benign samples and the adversarial ones were indistinguishable by the human eye [43]. Through this procedure, attackers aim to manipulate the DL/ML and cause it to perform false predictions [44]. Worryingly, this has severe implications with respect to adopting AI technologies in daily life applications. For instance, in the field of autonomous cars, most attacks target image-based classifiers, among which DL networks represent the most prominent approach since 2015 [45]. Here, adversaries may attack a self-driving car's system by prompting the underlying DL/ML model to ignore a 'stop' sign (e.g., due to the noise caused by placing pictures over the original stop sign or varying viewing conditions), consequently leading to a fatal crash [46]. Another critical field, in which adversarial samples may affect the comprehensive deployment of AI technologies in real-life applications is that of healthcare. Here, medical doctors and other related actors (such as insurance companies) rely on medical imaging systems to determine whether patients must undergo certain medical procedures. In order not to jeopardise patients' health and at the same time reduce the costs of treatment, concerned decision makers (medical doctors and insurance companies) try to minimise the number of unnecessary surgical procedures by deploying state-of-the-art DL systems, so as to analyse the relevant input data of patients, including, for example, dermoscopy images in dermatology, X-rays in radiology, and/or ophthalmic images, and make corresponding decisions [47,48]. Injecting adversarial samples into healthcare systems that use DL would result in an outcome that does not truly describe the underlying case and, hence, would have serious implications both for health and in terms of public and private costs.

### 3.3. Low Generalisation Capacity

Quite often, model-centric AI algorithms achieve high performance by utilising annotation artefacts. These are unnoticed patterns of context-free associations that come about through applying certain cognitive heuristics by human annotators [49,50]. Their effect, however, becomes obvious when the model is trained on a dataset that contains instances of such artefacts.

In natural language inference (NLI), for instance, large-scale datasets emerge from pairing a given sentence (premise) with three new ones (hypotheses), such that the given premise either entails, contradicts, or is logically neutral with respect to the generated hypothesis [49]. The hypotheses are generated by human subjects in a crowd-sourcing process. It has been observed that human annotators usually adopt cognitive strategies and

heuristics when authoring hypotheses, so as to save mental and time resources [51]. As a result, many of the NLI datasets contain annotation artefacts or biases; therefore, when used in training AI/ML algorithms, the underlying model performs surprisingly well in a dataset that contains these artefacts but fails to generalise to datasets that do not have similar artefacts. This low generalisation capacity of model-centric AI is due to the fact that the model has learned only the hypotheses (hypothesis-only bias [49,50]) but not the relationship between these hypotheses and their premises [52].

## 4. From Model-Centric AI to Data-Centric AI

Before AI researchers and ML practitioners began to deploy the term 'data-centric AI' in their scientific communications, they had years of practice in preparing datasets for training and testing their ML models. It can easily be assumed that substantial efforts were made at this stage of data preparation, in order to obtain high-quality data instances (data points). Yet, it frequently occurred that some instances were mislabelled, ambiguous, or fully irrelevant. Such data instances were considered invalid and hence had to be excluded from the datasets. However, in case the number of invalid data instances in a dataset is relatively small (in comparison with the dataset's size), keeping these points within the dataset would have a negligible impact on the performance of an ML model, and researchers could deal with this situation without fearing further consequences [53].

There are the reasons why businesses and industries may need to focus on ensuring high-quality datasets when building their AI systems, rather than dealing with this part of the development life cycle of the system as a one-time activity.

### 4.1. Limited Data

Above all, the lack of sufficiently large datasets that are comprehensive both in type and volume is especially paramount. Unlike Internet companies (such as Google and Baidu), manufacturing industries are often limited in the quantity of data they possess. Usually, they run their training models on datasets with $10^2$–$10^3$ relevant data points [54]. Thus, a model trained on no more than $10^3$ relevant instances to detect some fault (or a rare disease) would struggle when deploying techniques that were built for hundreds of millions of data points.

### 4.2. Solution Customisation

There is also a need for highly customised solutions. Consider, for example, a manufacturing business with more than one product. A single one-size-fits-all AI system for fault detection across all products would not work well, as each manufactured product would require a distinctly trained ML system.

### 4.3. Characteristics of Data-Centric AI

Data-centric AI embraces a continuous evaluation of the devised AI model in combination with data updates (Table 1). Typically, during the production stage, a devised AI model will be trained on a dataset only once before the process of software development can be ended with the deployment of the desired functionality (Figure 1A). However, because data-centric AI assumes successive improvements in data [17], in particular, in businesses and industries that cannot afford to have millions of data points (e.g., manufacturing, agriculture, and healthcare [20]), the underlying model (represented through the implemented algorithm) will inevitably access novel instances of data points that are completely different from those encountered during the training phase. Consequently, assessing the quality of the model, too, would recur more frequently rather than occurring only once. Furthermore, with the capacity of productions systems to provide timely feedback, this would result in a model (and, as a result, an AI system) being capable of recognising and, hence, reacting properly to distributional data drifts (if desired, this could serve as a prerequisite for online learning too [54]). Indeed, it is due to this faculty that the data-centric approach has an

edge over its model-centric counterpart when applied to models such as the one adopted in the steel inspection problem in [19] (Section 1).

### 4.4. Ongoing Growth of the IoT

According to *Statista*, a German research company specialised in market and consumer data, the number of worldwide IoT-connected devices would triple from 9.7 billion in 2020 to more than 29 billion in 2030.

A previous forecast, published in May 2020 by *Transforma Insights*, another leading research firm focused on the world of 'digital transformation', estimated the same figure to grow from 7.6 billion in 2019 to 24.1 billion in 2030 [55]. Later [56], however, *Transforma Insights* updated its estimate, confirming that of *Statista* (ostensibly due to taking into account the role of COVID-19 in accelerating the digital transformation, as the initial findings were published before enfolding the whole scope of the pandemic).

Not only do the analysts of both firms expect similar growth of IoT-connected devices over the next decade but they also agree on a compound annual growth rate (CAGR) of near to 10% of the IoT market, forecasting the global annual revenue of IoT to grow from USD 388 billion in 2019 to USD 1058.1 billion in 2030.

**Table 1.** Characteristics of model-centric AI and data-centric AI.

| Category | Model-Centric AI | Data-Centric AI | References |
|---|---|---|---|
| System development lifecycle | Successive upgrade of a model (algorithm/code) with fixed volume and type of data | Continuous improvement in the quality of data with fixed model hyperparameters | [19,20] |
| Performance | Performs well only with large datasets | Performs well also with smaller datasets | [37–39,54] |
| Robustness | Susceptible to adversarial samples | Higher adversarial robustness | [50,57] |
| Applicability | Appropriate for testing algorithmic solutions in applications with narrow tasks | Particularly suitable for real-world scenarios | [17,21] |
| Generalisation | Limited capacity to generalise across datasets (due to lack of context) | May generalise well to datasets other than the one tested on | [49,51,52,58] |

## 5. Rules and Criteria for Achieving Data-Centric AI

Adopting a data-centric approach can be achieved by combining several steps. Indeed, it is an activity for which empowering an AI system to accomplish the highest levels of performance requires incorporating interdisciplinary expertise. In the following, we consider some criteria and practical rules that help implement a data-centric approach leading to more effective AI systems.

### 5.1. Sufficient and Representative Data Inputs

First, datasets should be curated so as to reflect both sufficiency and representativeness of data inputs. One way to fulfil these requirements is by allowing the devised model to access as many data inputs with as much task-relevant information as required for the model to solve the task at hand. This includes cancelling out inevitable noise that is present in real life. Indeed, this is how the human brain applies 'selective attention' when processing visual information. Specifically, it focuses on the goal-relevant aspects of the environment while inhibiting distracting information that might be otherwise noisy for the task at hand [59].

For research teams, this can practically mean, for example, reducing the spatial complexity in an image segmentation task to the relevant image regions rather than keeping on tuning the devised model architecture, model complexity, data augmentation strategies, or related training strategies [60].

### 5.2. Unveiling Inherent Contexts through Textual Descriptions

Second, research teams should work towards ensuring high-quality data by revealing the inherent context within data inputs during data preparation. A vital *human* activity during the process of data preparation is that of 'data labelling' (also known as 'data annotation'). Research teams usually overlook potential biases, which may have a detrimental effect on the quality of data.

Specifically, data annotators are humans with different cultural and individual backgrounds. Thus, they are susceptible to subjective judgements that could result in erroneous labels. To avert falling into the trap of biases, however, research teams may require including 'textual descriptions' as an intermediate step between accessing data inputs (images, videos, audio recordings, etc.) and assigning labels to them. Such textual descriptions would be some 3- to 10-word sentences in which human subjects (e.g., annotators) reflect on whatever contextual information is included in the data inputs. Although this admittedly increases the overall time required to create datasets, it helps AI engineers ensure that the collected data clearly illustrate the concepts that they need the AI to learn, resulting in AI systems that are able to learn from smaller datasets available in most industries.

### 5.3. Continuous Involvement of AI and Business-Domain Experts

Third, and closely related to the step above, data engineering should be performed by business-domain experts instead of AI experts. This is because AI experts have their competency mainly in representing the world in a format that enables the machine (algorithm) to learn patterns, while domain experts have comprehensive knowledge about the intricacies of a specific business use case and can hence provide a domain-relevant representation of the world [61]. Moreover, domain experts can play a role in enhancing the evaluation process by developing specific use cases that put the model into more domain-sensitive tests. This way, the ability to use AI becomes easier, and hence utilised AI system becomes more accessible to a wide range of industries.

### 5.4. Use of MLOps

Fourth, instead of spending time and effort on developing software, research teams and experts could reduce the maintenance cost of AI applications by using machine learning operation (MLOp) platforms. These provide much of the scaffolding software needed to facilitate the production of an AI system. As a result, the gap between proof of concept and production will massively shrink to weeks, instead of otherwise years.

There are several MLOp systems that could be applied for both data-centric and model-centric AI. For instance, one may use tools such as labelme [62] and ActiveClean [63] for data-labelling and data-cleaning activities, respectively.

In the case of model-centric AI, examples of available MLOp tools include model store systems [64], model continuous integration tools [65,66], training platforms [67], and deployment platforms [68].

## 6. Reconciling Data-Centric AI with Model-Centric AI
### 6.1. Data-Centric and Model-Centric AI Can Only Be Two Sides of One Coin

At first glance, it seems quite natural to think of tweaking the model (algorithm/code) rather than improving the data, if the goal is to enhance the system's performance and robustness. However, with the limitations of model-centric AI solutions and the serious problems DL-based models suffer from, as described in Section 3 (or Table 1 for summary), the need to turn to data-centric approaches becomes quite understandable. We believe that this should occur while paying similar attention to the underlying model; that is, both

model-centric and data-centric approaches should be handled as two sides of the same coin, and recognising the limitations of the model-centric approach in certain businesses and industries should not lead to abandoning it altogether. This is both practical and self-evident since both the model and data affect each other interdependently.

Moreover, though shifting to data-centric AI approaches in response to the limitations of model-centric AI might seem attractive, constructing new datasets usually comes with high costs, and there is no guarantee that the new dataset would be completely free from new types of artefacts [52,69]. Indeed, due to the brain's flexibility, there is always a possibility for human annotators to apply subtle cognitive strategies during the annotation task that would limit the model's ability to generalise. So, it would be better to treat the model and data as two sides of one coin and improve them simultaneously.

### 6.2. Problem-Solving Requires Considering Both the How-To (Model) and the What-Is (Data)

Our intuitive understanding of problem solving puts considerable weight on the way we act upon things besides knowing their properties and facts about them. In the case of the development of AI systems, this would imply a preference towards exerting more efforts in the design and optimisation of intelligent algorithms and more vividly tweaking the underlying model hyperparameters along with the code used to implement the designed algorithms. Keeping the dataset fixed at this stage of work is even necessary since only through a fixed dataset can one have the possibility to compare models and classify them according to their performance.

On the other hand, starting the design of AI systems with a model-centric approach provides ample opportunity to accumulate experience in understanding real-world problems and potential computational solutions compared with the gains in the experience that could have resulted, had research and industry followed the track of data-centric approach. In fact, it is in the nature of things that when solving a problem, we first take action in order to produce an effect, upon which we then proceed to dig deeper to understand the properties of things around us. This is exactly what intelligent agents do to acquire experience while interacting with their environments, as postulated by reinforcement learning [70], a recently much-celebrated ML technique [26,58,71,72].

### 6.3. Models' Limitations Do Not Necessarily Imply the Limitation of Modelling

According to Aristotle's first-principles thinking, understanding the fundamental aspects of a problem can help provide good solutions to it [73]. In the NLI task, which was introduced in Section 3.3, the model learns, among other things, annotation artefacts. This results in high performance when tested on datasets containing such artefacts but not on ones with no such biases.

While this shows the limitations of the devised model, it does not necessarily mean that the model-centric approach itself is limited (at least in the field of NLI). Specifically, by understanding that the annotation artefacts result from ignoring the premise–hypothesis relationship, one can design models that suppress the learning of such artefacts. For instance, Belinkov et al. ([52]) proposed to input into the model both the hypothesis and entailment label so as to predict the premise instead of the typical NLI models, which learn to predict an entailment label, given a premise–hypothesis pair.

### 6.4. Nature Supports Learning of Models

Several strategies contribute to enhancing the robustness of DL/ML models and paving the way for a complementary interplay between data-centric and model-centric AI. First and foremost is aligning algorithms to nature with its intuitive and robust mechanisms, in particular, learning and 'problem-solving' mechanisms as performed by the human brain. One such mechanism is *common sense*. It describes the ability of humans (as well as animals) to learn *world models* by accumulating enormous amounts of *contextual* knowledge about how the world works and what causal relationships regulate the co-occurrence and succession of events, and use this knowledge in predicting future outcomes [74–76].

Typically, common-sense knowledge is acquired through unsupervised observation in a very limited number of task-irrelevant interactions with the learning environment.

Current AI and ML systems do not exhibit such a faculty. For instance, while a human car driver can draw on her/his intuitive knowledge of physics to predict the bad consequence of driving too fast, an AI system of an autonomous vehicle requires thousands of reinforcement learning trials to acquire such knowledge. A typical observation in this regard is that learning occurs at a meta-level; that is, when humans solve problems that they face for the first time, they typically rely on the affordances the brain provides, sometimes fleetingly, through interaction with the environment (see [74,77]).

### 6.5. Together, Data-Centric AI and Model-Centric AI Provide More Robustness and Security

The fact that data could be freely submitted into a running DL/ML algorithm (as can be experienced in cyber attacks with adversarial samples) without directly attacking the model itself or the IT infrastructure that hosts it provides a supportive argument for the significance of developing data-centric and model-centric AI in a complementary approach. Initially, it was believed that the tendency of a DL network to make false predictions in response to the inputs formed by applying small but intentionally worst-case perturbations to samples from the dataset was due to the extreme nonlinearity of DL networks, combined with insufficient model averaging and the insufficient regularisation of the purely supervised learning problem. Later, however, it was understood that adversarial samples are manifestations of the rather linear nature of neural networks in general. They are neither random artefacts of the normal variability that accumulates through different runs of propagation learning nor do they arise from overfitting or due to incomplete model training [48]. Rather, adversarial samples are robust to random noise and as such could transfer from one neural network model to another, despite differences in the number of layers with different model hyperparameters and most importantly, when trained on different sets of samples [41].

This implies that rather than being solely a matter of the datasets used to train a deep neural network model based on backpropagation, it is the way how the network is structurally connected to data distribution that matters, suggesting combining both data-centric and model-centric approaches when aiming at enhancing network robustness.

### 7. Conclusions

In response to the increasing interest among the AI/ML community in favoring data-centic AI over model-centric AI, which suggests a paradigm shift in the way AI/ML models are devised, we analysed the pros and cons of both approaches, only to end up reconciling data-centric AI with model-centric AI. On the one hand, we share other researchers' view that more attention should be paid to the data-centric AI approach. However, we believe that this should not result in reduced interest in model-centric AI, as promoting current AI requires considering both approaches in a complementary interplay. In particular, Industry 4.0, which promises to automate the interaction of cyber–physical objects over the IoT, would benefit from such an interplay, as both data and AI are hallmarks of Industry 4.0 technologies.

**Funding:** This research received no external funding.

**Institutional Review Board Statement:** Not applicable

**Informed Consent Statement:** Not applicable

**Data Availability Statement:** Not applicable

**Conflicts of Interest:** The author declares no conflict of interest.

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
