# Peer review of "Data-Centric and Model-Centric AI: Twin Drivers of Compact and Robust Industry 4.0 Solutions"

_applsci, doi:10.3390/app13052753_

Round 1

Reviewer 1 Report

 Thanks to the author's meticulous work, I think this paper has reached the published level.

Author Response

Dear Anonymous Reviewer 1, 

Thank you for having the time reviewing and evaluating my work. I have included my responses in the uploaded word document. 

Sincerely, 

Reviewer 2 Report

Thank you for giving me the opportunity to learn from your research.

I think the article provides a good overview of the status quo of model-centric and data-centric AI approaches. The literature used and the core statements are comprehensively and solidly compiled.

However, my expectation of the outcome of the paper was somewhat higher after reading the abstract and section 1.1 (our contribution). It is more of a descritical overview article, less of a truly analytical comparison of two AI approaches. It is announced that in section 3.2 current AI will be connected to the field of cybersecurity by showing, through the phenomenon of adversarial samples, the limitations of model-centric AI in terms of algorithmic stability and robustness. This is also done in a relatively concise chapter, but most of the statements refer to the state of the literature; it remains unclear what the research group or the author has specifically investigated and developed. In any case, it is always referred to as our work and our contribution, the article and at least one reference to previous work only refers to the author himself. The article should elaborate more on what is a description of the state of the literature and what the author has produced in terms of his own research work.  

Thank you.

Author Response

Dear Anonymous Reviewer 2, 

Thank you for having the time reviewing and evaluating my work. I have included my responses in the uploaded word document. 

Sincerely,

Reviewer 3 Report

§  In introduction the reasons for change from model-centric to data centric approach could be explained in detail, not just providing two references regarding large data sets and highly customized solutions.

§  The reason why the authors support “both and” approach could be explained in detail.

§  There is no specific scientific method selected and applied in the article. The scientific method for comparing model-centric and data-centric AI approaches would involve developing or selecting appropriate models and data sets for each approach, training and testing the AI systems using established evaluation metrics and comparing the results to determine which approach performs better or has certain advantages or disadvantages. Additionally, it would be important to consider the interpretability, explainability and generalization capabilities of the AI models.

§  The comparison of model-centric AI vs data-centric AI approaches could be presented in a Table including all the attributes.

§  There is no argument regarding how data-centric AI and model-centric AI contribute to Industry 4.0 solutions.

Author Response

Dear Anonymous Reviewer 3, 

Thank you for having the time reviewing and evaluating my work. I have included my responses in the uploaded word document. 

Sincerely,

Round 2

Reviewer 2 Report

Thank you for giving me the opportunity to read the revised version.

Reviewer 3 Report

After revision, the originality, importance, value added of the paper and potential contribution to the journal has been improved. 

The revised paper has been well articulated dealing with sufficiently new and original concepts. The reasons for change from model-centric to data centric approach are clearer now. Method is now clearer. The comparison has been presented clearly in a Table. The link of data-centric AI and model-centric AI to Industry 4.0 is clearer now.